

# Assessment of eight insulin resistance surrogate indexes for predicting metabolic syndrome and hypertension in Thai law enforcement officers

Rujikorn Rattanatham[1,2], Jitbanjong Tangpong[1,2], Moragot Chatatikun[1,3], Dali Sun[4], Fumitaka Kawakami[5,6], Motoki Imai[5,7] and Wiyada Kwanhian Klangbud[1,3,8]

[1] School of Allied Health Sciences, Walailak University, Thasala, Nakhon Si Thammarat, Thailand
[2] Research Excellence Center for Innovation and Health Products, Walailak University, Thasala, Nakhon Si Thammarat, Thailand
[3] Center of Excellence Research for Melioidosis and Microorganisms (CERMM), Walailak University, Thasala, Nakhon Si Thammarat, Thailand
[4] Biomedical Engineering Program, North Dakota State University, Fargo, North Dakota, United States
[5] Research Facility of Regenerative Medicine and Cell Design, School of Allied Health Sciences, Kitasato University, Sagamihara, Kitasato, Japan
[6] Department of Regulation Biochemistry, Kitasato University Graduate School of Medical Sciences, Sagamihara, Kitasato, Japan
[7] Department of Molecular Diagnostics, School of Allied Heath Sciences, Kitasato University, Sagamihara, Kitasato, Japan
[8] Walailak University Medical Technology Clinic (Wu-MeT), Walailak University, Thasala, Nakhon Si Thammarat, Thailand

Corresponding author
Wiyada Kwanhian Klangbud, wiyadakwanhian@gmail.com

## ABSTRACT

Police officers in Thailand have an increased risk of heart disease, stroke, and type 2 diabetes, possibly due to a high prevalence of hypertension and metabolic syndrome (MetS). In this study, the researchers aimed to understand the relationship between surrogate markers of insulin resistance (IR) and the prevalence of MetS and hypertension in Thai police officers. The study included 7,852 police officer participants, of which 91.8% were men with an average age of 48.56 years. The prevalence of hypertension and MetS were found to be 51.1% and 30.8%, respectively, and the participants with MetS and hypertension were older compared to the regular group. The study looked at eight IR indices, including markers such as atherogenic index of plasma (AIP), lipid accumulation product (LAP), metabolic score for insulin resistance (METS-IR), triglyceride glucose (TyG) index, TyG index with body mass index (TyG-BMI), TyG index with waist circumference (TyG-WC), the ratio of triglycerides to high-density lipoprotein cholesterol ratio (TG/HDL-c), and visceral obesity index (VAI). These indices were found to be positively correlated with waist circumference, systolic blood pressure (SBP), diastolic blood pressure (DBP), fasting plasma glucose (FPG), and triglycerides (TG), while being negatively correlated with high-density lipoprotein cholesterol (HDL-c). In addition, the multiple regression analysis showed that higher quartiles of all IR indices were significantly associated with increased risks of MetS and hypertension. Interestingly, the IR indices were more accurate in predicting MetS (ranges 0.848 to 0.892) than traditional obesity indices, with the AUC difference at $p < 0.001$. Among the IR

indices, TyG-WC performed the best in predicting MetS (AUC value 0.892 and Youden index 0.620). At the same time, TyG-BMI had the highest accuracy in predicting hypertension (AUC value of 0.659 and Youden index of 0.236).

In addition, this study found that when two markers were combined for diagnosing metabolic syndrome, a significantly improved predictive value for disease risk was observed, as evidenced by higher AUC and Yoden index. Moreover, the IR indices were found to have higher predictive power for MetS and hypertension in younger police personnel (age < 48 years) than older personnel. In conclusion, this study highlights the importance of reducing cardiovascular disease risks among law enforcement personnel as a strategic goal to improve their health and wellness. The findings suggest that IR indices may be valuable tools in predicting MetS and hypertension in law enforcement personnel and could potentially aid in the early identification and prevention of law enforcement personnel health conditions.

## INTRODUCTION

Metabolic syndrome (MetS) is a cluster of metabolic conditions that increase the risk of cardiovascular diseases and type 2 diabetes mellitus. MetS is a cluster of symptoms that includes an abdominal pattern of obesity with an increased waist circumference, dyslipidemia characterized by higher serum triglyceride and low HDL cholesterol, elevated blood pressure, and impaired glucose tolerance. The location and distribution of fat inside the body are excellent indicators of MetS and its related risk factors. BMI and WC have been widely used to assess central obesity and MetS for a long time. However, BMI and WC alone are ineffective for estimating central obesity and predicting cardiometabolic risk (*Elagizi et al., 2018*). Therefore, a reliable anthropometric indicator for visceral and abdominal obesity, which give simple, inexpensive, and effective predictors of metabolic health issues, is essential. Thus, the indices calculated from general parameters performed in a routine test were generated.

Hypertension, one of the most critical risk factors for cardio-cerebrovascular disease, renal dysfunction, and cognitive impairment, affects millions of people and is the leading cause of disability and death globally (*Han et al., 2021*). According to clinical studies, hypertension patients are usually observed to coexist with type 2 diabetes mellitus. Furthermore, according to substantial evidence, insulin resistance plays a crucial role in the onset of hypertension (*Mancusi et al., 2020*). Therefore, the level of insulin resistance could be utilized to predict the occurrence of hypertension.

The hyperinsulinemic-euglycemic clamp technique is the most popular direct approach for evaluating insulin resistance. However, it is invasive, complicated, and impractical (*Tam et al., 2012*). The homeostasis model assessment for insulin resistance (HOMA-IR) index, the most popular indirect technique, is susceptible to the precision of insulin measurement and has low consistency (*Luo et al., 2022*). BMI and WC have been widely

used to assess central obesity and MetS for a long time. However, BMI and WC alone are ineffective for estimating central obesity and predicting cardiometabolic risk (*Elagizi et al., 2018*). Therefore, it is imperative to conduct research to identify IR surrogate markers that are less complicated, more precise, and more practically applicable for predicting hypertension and MetS, which can effectively mitigate the risk of hypertension and MetS among police personnel. Several IR surrogates have been developed, which some simple routine biochemical indicators can calculate. Surrogate markers for evaluating insulin resistance include triglyceride glucose (TyG) index, TyG index with body mass index (TyG-BMI), TyG index with waist circumference (TyG-WC) (*Song et al., 2022*), the ratio of triglycerides to high-density lipoprotein cholesterol ratio (TG/HDL-c) (*Aslan Çin et al., 2020*; *Zhang et al., 2021*), the metabolic score for insulin resistance (METS-IR) (*Bello-Chavolla et al., 2018a*). Lipid accumulation product (LAP) and Visceral obesity index (VAI) are predictors of cardiovascular (*Zhao et al., 2021*), cerebrovascular risks (*Zhang et al., 2022*) are considered clinical indicators of MetS (*Huang et al., 2022*; *Jiang et al., 2022*). LAP is calculated by triglyceride and waist circumference. VAI is computed by integrating anthropometric data and metabolic parameters. Triglycerides and high-density lipoprotein cholesterol are components of the atherogenic index of plasma (AIP). It is a new marker for evaluating atherogenicity risk and cardiometabolic status (*Khosravi et al., 2022*).

The law enforcement officer is a high-stress vocation associated with higher cardiovascular disease prevalence and mortality risk (*Magnavita et al., 2018*). The demanding nature of law enforcement work puts officers at an increased risk of metabolic syndrome. The high-stress levels and irregular work schedules that police officers often face can lead to poor dietary choices, lack of physical activity, and disrupted sleep patterns (*Yates et al., 2021*). Additionally, law enforcement officers are more likely to engage in cigarette and alcohol usage, prolonged duty hours and frequent night shifts result in continuous secretion of catecholamine, leading to elevated blood pressure and MetS (*Chauhan et al., 2022*). Police personnel reportedly have a high prevalence of hypertension and MetS, which further contributes to their health deterioration and unavailability for duty (*Yates et al., 2021*). Recent studies conducted in Thailand have shown that law enforcement officers have a higher prevalence of MetS compared to the general population, with rising rates of overweight or obesity and associated hypertension among military personnel (*Gurung et al., 2023*; *Napradit et al., 2007*). Hence, prioritizing the health and well-being of law enforcement officers is crucial. Regular health screenings, early detection, and management of risk factors through appropriate medical interventions can effectively reduce the risk of MetS and hypertension among law enforcement officers.

This study aims to investigate the relationships between eight IR surrogates (AIP, LAP, METS-IR, TG/HDL-c, TyG index, TyG-BMI, TyG-WC, and VAI) and the prevalence of MetS and hypertension in Thai police officers, as well as to compare the effectiveness of IR surrogate indices and conventional indices in identifying hypertension and MetS.

## MATERIALS AND METHODS

### Data collection and sample

This cross-sectional study was conducted in 2019 and enrolled individuals who underwent annual health examinations at 166 police stations in nine provinces in southern Thailand. Participants were included if they were aged 18 years or above, of both genders and free from severe chronic diseases such as hepatic and kidney diseases. However, 3,666 out of the initial 13,688 participants were excluded due to incomplete biochemical information, including fasting plasma glucose (FPG), triglyceride (TG), low-density lipoprotein-cholesterol (LDL-C), high-density lipoprotein cholesterol (HDL-C), and total cholesterol (TC). Additionally, anthropometric data, such as age, sex, waist circumference (WC), weight, height, systolic blood pressure (SBP), diastolic blood pressure (DBP), heart rate, body mass index (BMI), and medication history of using antihyperglycemic or antihypertensive drugs were missing for 2,168 participants. As a result, these participants were excluded from the analysis. Finally, a total of 7,852 participants were included in the study. The study protocol was reviewed and approved by the Walailak University Ethics Committee for Human Research (approval no. WUEC-21-349-01). The documentation of informed consent was waived by the ethics committee. All of the data and code were in File S1.

### The demographic data and anthropometric measurements

The demographic data and anthropometric measurements were obtained, including age, sex, WC, weight, height, systolic blood pressure (SBP), diastolic blood pressure (DBP), heart rate, and medication history. In addition, the body mass index (BMI) was calculated. Blood pressure was assessed on the participant's right arm while seated, following a minimum of 10 min of rest, utilizing a standard mercury sphygmomanometer. The average of two readings was recorded as the individual's blood pressure. Blood samples were analyzed for fasting plasma glucose (FPG), triglyceride (TG), low-density lipoprotein-cholesterol (LDL-c), high-density lipoprotein-cholesterol (HDL-c), and total cholesterol (TC) after at least 8 h of overnight fasting. Lipid profiles, including total cholesterol, triglycerides, HDL-c, and LDL-c were measured using Mindray kits (Mindray, Shenzhen, China). The total cholesterol kit utilized the cholesterol oxidase-peroxidase (CHOD-POD) method, where absorbency increased proportionately with cholesterol levels. Triglyceride levels were measured using the glycerokinase peroxidase-peroxidase (GPO-POD) method. Principle of direct method for both HDL-c and LDL-c. Glucose levels were detected using the Glucose Kit (Mindray, Shenzhen, China) based on the glucose oxidase-peroxidase (GOD-POD) method, with the glucose concentration directly proportional to the quinoneimine dye. Hypertension was defined as the presence of at least one of the following conditions: SBP ≥140 mmHg or diastolic blood pressure (DBP) ≥90 mmHg or using antihypertensive drugs. Raised WC in the Asian population was defined by males with WC >90 cm and males with WC >80 cm. MetS was indicated when three or more of the following five criteria were met: (1) abdominal obesity (WC ≥90 cm in males

and ≥80 cm in females), (2) TG ≥1.7 mmol/L, (3) HDL-c <1.03 mmol/L in males and <1.29 mmol/L in females, (4) SBP ≥130 mmHg or DBP ≥85 mmHg, and (5) FPG ≥5.6 mmol/L.

The IR surrogate indicators were calculated using the following formula (*Cheng, Kong & Chen, 2022*; *Kahaer et al., 2022*; *Sheng et al., 2021*):

TyG index = log (fasting TG × FPG/2)

TyG-BMI = TyG × BMI (*Er et al., 2016*)

TyG-WC = TyG × WC (*Sheng et al., 2021*)

TG/HDL-c = TG/HDL-c (*Abbasi & Reaven, 2011*)

METS-IR = ln [2 × FPG + TG × BMI/ln [HDL-c] (*Bello-Chavolla et al., 2018b*)

LAP (men) = WC − 65 × TG

LAP (women) = WC − 58 × TG (*Kahn, 2005*).

VAI (man) = [WC/39.68 + (1.88 × BMI)] × (TG/1.03) × (1.31/HDL-c);

VAI (women) = [WC/36.58 + 1.89 × (BMI)] × (TG/0.81) × (1.52/HDL-c) (*Jiang et al., 2022*).

AIP = log (TG/HDL-c)

## Statistical analysis

Statistical analyses were performed with SPSS version 26 (SPSS Inc., Chicago, IL, U.S.). An assessment of the normality of the continuous data uses Kolmogorov–Smirnov test, skewness and kurtosis. Histograms and the absolute skewness and kurtosis values are used to determine the normality of data samples larger than 300. Therefore, either an absolute skewness value of ≤2 or an absolute kurtosis of ≤4 may be utilized as reference values for establishing substantial normality (*Mishra et al., 2019*).

Continuous variables with normal distribution were presented as the mean and standard deviation. The variables with skewed distribution were shown as the median and interquartile range (IQR). Categorical variables were described as numbers and percentages. Two continuous variables were compared using the Student's t-test (normal distribution) and the Mann–Whitney U test (skewed distribution). More than two continuous variables were compared using ANOVA. The Kruskal-Wallis test is the non-parametric alternative to the one-way ANOVA. The Chi-square test was used to compare categorical variables. Correlations between IR surrogate indices and metabolic components were assessed using Pearson's (for continuous variables with normal distribution) and Spearman's (for continuous variables with skewed distribution and categorical variables) methods. Logistic regression was used to analyze the relationship between the various IR indices and the risk of MetS and hypertension. A receiver operating characteristic (ROC) curve was used to calculate the area under the curve (AUCs) and assess the predictive efficacy of IR surrogates for MetS and hypertension. The MedCalc program was used to obtain AUCs. Youden's index was used to identify the optimal cut-off point, calculated based on each IR surrogate's corresponding sensitivity and specificity (*Barrett & Fardy, 2021*). The level of statistical significance was accepted at the two-sided 0.05 level, and the confidence interval (CI) was determined at the 95% level.

**Table 1 Baseline characteristics of participants based on metabolic syndrome and hypertension.**

| Parameter | Non-MetS n = 5,431 | MetS n = 2,421 | p-value | Normotension n = 3,841 | Hypertension n = 4,011 | p-value |
|---|---|---|---|---|---|---|
| Male (%) | 4,867 (89.6) | 2,291 (94.6) | <0.001 | 3,335 (86.8) | 3,823 (95.3) | <0.001 |
| Age (years) | 48.31 ± 6.51 | 49.11 ± 6.18 | <0.001 | 47.72 ± 6.61 | 49.6 ± 6.12 | <0.001 |
| BMI (kg/m²) | 23.89 ± 2.80 | 26.52 ± 3.43 | <0.001 | 23.96 ±2.99 | 25.41 ± 3.33 | <0.001 |
| WC (cm) | 82.14 ± 6.02 | 88.57 ± 8.01 | <0.001 | 82.66 ± 7.05 | 85.53 ± 7.30 | <0.001 |
| SBP (mmHg) | 130.30 ± 15.90 | 140.40 ± 15.89 | <0.001 | 122.21 ± 10.00 | 144.15 ± 14.36 | <0.001 |
| DBP (mmHg) | 86.30 ± 11.67 | 93.04 ± 11.04 | <0.001 | 79.66 ± 6.52 | 96.76 ± 9.65 | <0.001 |
| Weight (kg) | 67.40 ± 9.25 | 75.89 ± 10.81 | <0.001 | 67.54 ± 9.94 | 72.39 ± 10.51 | <0.001 |
| Height (cm) | 167.86 ± 5.93 | 169.09 ± 5.66 | <0.001 | 167.73 ± 6.13 | 168.72 ± 5.57 | <0.001 |
| FPG (mmol/L) | 5.00 (4.29−5.71) | 5.72 (4.29−7.15) | <0.001 | 5.00 (4.17−5.83) | 5.28 (4.23−6.33) | <0.001 |
| TC (mmol/L) | 5.46 (4.14−6.78) | 5.69 (4.14−7.24) | <0.001 | 5.50 ± 1.04 | 5.64 ± 1.15 | <0.001 |
| TG (mmol/L) | 1.23 (0.53−1.93) | 2.27 (0.99−3.55) | <0.001 | 1.32 (0.37−2.27) | 1.66 (0.42−2.90) | <0.001 |
| LDL-C (mmol/L) | 3.43 ± 0.98 | 3.53 ± 1.18 | <0.001 | 3.44 ± 0.99 | 3.48 ± 1.10 | <0.001 |
| HDL-C (mmol/L) | 1.43 ± 0.35 | 1.09 ± 0.28 | <0.001 | 1.36 ± 0.37 | 1.30 ± 0.36 | <0.001 |
| TyG index | 8.55 ± 0.48 | 9.38 ± 0.58 | <0.001 | 8.66 ± 0.60 | 8.95 ± 0.65 | <0.001 |
| TG/HDL-C | 0.89 (0.17−1.61) | 2.18 (0.52−3.84) | <0.001 | 1.00 (0.00−2.05) | 1.33 (0.00−2.72) | <0.001 |
| TyG-BMI | 204.45 ± 28.25 | 248.49 ± 33.92 | <0.001 | 207.87 ± 32.77 | 227.76 ± 36.92 | <0.001 |
| TyG-WC | 702.74 ± 69.32 | 829.94 ± 84.36 | <0.001 | 716.70 ± 88.92 | 766.15 ± 93.75 | <0.001 |
| METS-IR | 34.21 (26.95−41.47) | 43.25 (34.90−51.60) | <0.001 | 35.59 ± 6.58 | 39.20 ± 9.66 | <0.001 |
| LAP | 21.85 (5.07−38.63) | 51.76 (14.81−88.71) | <0.001 | 30.80 ± 26.74 | 43.61 ± 38.54 | <0.001 |
| VAI | 1.15 (0.25−2.05) | 2.80 (0.75−4.85) | <0.001 | 1.79 ± 1.74 | 2.31 ± 2.20 | <0.001 |
| AIP | −0.05 (−0.39−0.29) | 0.34 (0.02−0.66) | <0.001 | 0.02 ± 0.31 | 0.14 ± 0.32 | <0.001 |

**Note:**
Value is shown as median (interquartile range; IQR).

## RESULTS

Among the 7,852 participants, 7,158 (91.2%) were male, and 694 (8.8%) were female. The average age of the entire population was 48.56 ± 6.42 years. The prevalence of MetS and hypertension was 30.8% and 51.1%, respectively. The clinical characteristics of the study population are summarized in Table 1. Participants with MetS were older than those without MetS, and patients with hypertension were older than those with normotension. The mean values of BMI, WC, SBP, DBP, FPG, TC, TG, LDL-c, HDL-c, AIP, LAP, METS-IR, TG/HDL-c, TyG index, TyG-BMI, TyG-WC, and VAI were significantly higher in hypertensive patients compared to normotensive participants, and in the MetS group compared to the non-MetS group (all $p < 0.001$) (Table 1).

### Correlation between MetS, hypertension, and IR surrogates

The correlation coefficients between the IR surrogate indices and MetS components are shown in Fig. 1. Insulin surrogate indices AIP, LAP, METS-IR, TG/HDL-c, TyG index, TyG-BMI, TyG-WC and VAI correlated positively with WC, SBP, DBP, FPG, and TG, but negatively with HDL-c.

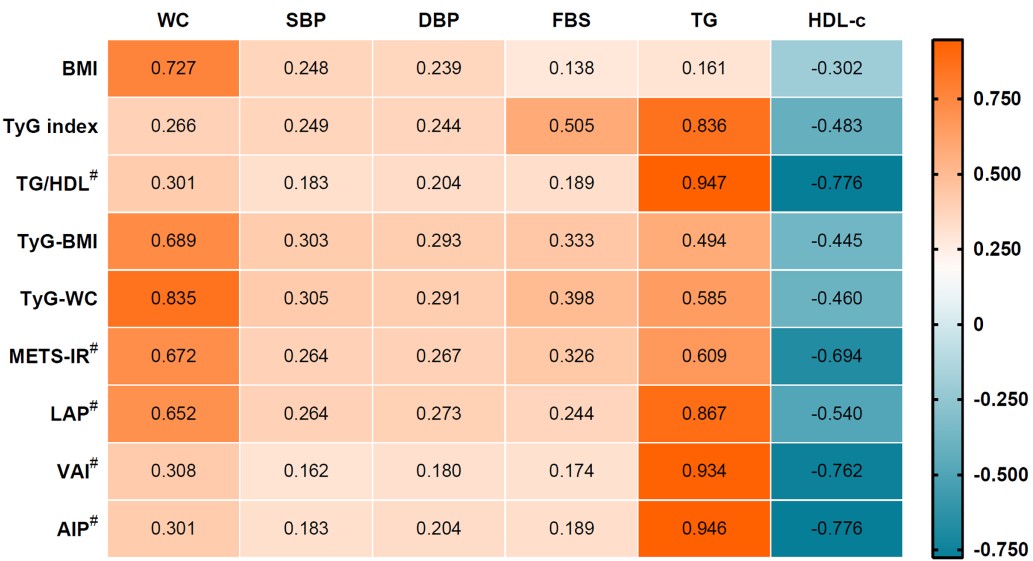

**Figure 1 Correlation between insulin resistance surrogate and metabolic component.** #For the non-parametric used Spearman's correlation. All correlation results had a $p$-value < 0.001.

## Multiple logistic regression analyses of the IR surrogates and the prevalence of MetS and hypertension

All IR surrogates were divided into four quartiles, with the lowest quartile as a reference group. The prevalence of hypertension and MetS increased significantly with the elevated quartile of IR surrogate indices, AIP, LAP, METS-IR, TG/HDL-c, TyG index, TyG-BMI, TyG-WC and VAI. The relationship between each IR surrogate and MetS and hypertension was investigated using multivariate logistic regression. There was no covariate adjustment in the crude model; age and sex were adjusted as in Model 1; age, sex, FPG, BMI, and WC were adjusted in Model 2 for hypertension; age, sex, SBP, DBP, FPG, BMI, and WC were fully adjusted in Model 2 for MetS. After adjusting for all covariates in Model 2, compared with the first quartile (Q1), the other three quartiles of IR surrogates were strongly associated with MetS (all $p < 0.001$) and hypertension (all $p < 0.001$), as shown in Tables 2–3. Therefore, it indicated a higher risk for MetS and hypertension in the upper quartile of the AIP, LAP, METS-IR, TG/HDL-c, TyG index, TyG-BMI, TyG-WC and VAI, when compared with the reference group (Q1).

## Predictive efficacy of IR surrogate for MetS and hypertension prevalence

The ROC curve of different IR surrogates for MetS is shown in Fig. 2, and hypertension is presented in Fig. 3. The area under the ROC curve with its 95% CI for predicting hypertension and MetS by surrogates index of insulin resistance is shown in Table 4. This research revealed that the AIP, LAP, METS-IR, TG/HDL-c, TyG index, TyG-BMI, TyG-WC and VAI could all be used to identify the individuals who had MetS. TyG-WC showed the highest AUC value of 0.892 (95% CI [0.885–0.899]), with a cut-off value of 760.77 according to the highest Youden index of 0.620. Simultaneously, the AUC values

**Table 2 Multivariate logistic regression of different indices for metabolic syndrome.**

| Parameter | Crude model OR (95% CI) | p-value | Model 1 OR (95% CI) | p-value | Model 2 OR (95% CI) | p-value |
|---|---|---|---|---|---|---|
| **TyG index** | | | | | | |
| Q1; <8.35 | 1 | | 1 | | 1 | |
| Q2; 8.35–8.73 | 3.73 [2.69–5.16] | <0.001 | 4.07 [2.93–5.66] | <0.001 | 3.04 [2.13–4.33] | <0.001 |
| Q3; 8.74–9.16 | 23.67 [17.58–31.87] | <0.001 | 27.44 [20.20–37.27] | <0.001 | 22.19 [15.94–30.89] | <0.001 |
| Q4; >9.16 | 112.58 [83.29–152.15] | <0.001 | 133.59 [97.77–182.54] | <0.001 | 106.01 [75.14–149.56] | <0.001 |
| **TG/HDL** | | | | | | |
| Q1; <0.71 | 1 | | 1 | | 1 | |
| Q2; 0.71–1.14 | 3.58 [2.69–4.75] | <0.001 | 3.94 [2.95–5.25] | <0.001 | 2.99 [2.12–4.20] | <0.001 |
| Q3; 1.15–1.93 | 15.78 [12.14–20.51] | <0.001 | 18.55 [14.14–24.32] | <0.001 | 20.07 [14.51–27.76] | <0.001 |
| Q4; >1.93 | 77.68 [59.61–101.24] | <0.001 | 97.45 [73.85–128.58] | <0.001 | 139.74 [99.66–195.93] | <0.001 |
| **TyG-BMI** | | | | | | |
| Q1; <192.70 | 1 | | 1 | | 1 | |
| Q2;192.70–214.75 | 4.43 [3.37–5.82] | <0.001 | 4.36 [3.32–5.74] | <0.001 | 3.07 [2.32–4.07] | <0.001 |
| Q3; 214.76–239.42 | 15.84 [12.25–20.49] | <0.001 | 15.68 [12.10–20.32] | <0.001 | 8.52 [6.49–11.18] | <0.001 |
| Q4; >239.42 | 62.06 [47.91–80.38] | <0.001 | 61.95 [47.74–80.39] | <0.001 | 20.88 [15.65–27.85] | <0.001 |
| **TyG-WC** | | | | | | |
| Q1; <678.30 | 1 | | 1 | | 1 | |
| Q2; 678.30–733.92 | 7.95 [5.40–11.72] | <0.001 | 10.13 [6.79–15.10] | <0.001 | 10.50 [6.97–15.82] | <0.001 |
| Q3; 733.93–794.89 | 31.96 [22.01–46.38] | <0.001 | 43.55 [29.41–64.48] | <0.001 | 45.36 [30.01–68.57] | <0.001 |
| Q4; <794.89 | 224.08 [153.86–326.34] | <0.001 | 307.96 [207.07–457.98] | <0.001 | 337.40 [214.24–531.36] | <0.001 |
| **METS-IR** | | | | | | |
| Q1; <32.16 | 1 | | 1 | | 1 | |
| Q2; 32.16–36.69 | 3.98 [2.99–5.29] | <0.001 | 4.02 [3.02–5.34] | <0.001 | 2.78 [2.06–3.74] | <0.001 |
| Q3; 36.70–41.66 | 15.12 [11.59–19.73] | <0.001 | 15.44 [11.80–20.21] | <0.001 | 9.50 [7.15–12.62] | <0.001 |
| Q4; >41.66 | 86.95 [66.44–113.78] | <0.001 | 90.71 [69.08–119.11] | <0.001 | 40.25 [29.88–54.22] | <0.001 |
| **LAP** | | | | | | |
| Q1; <17.78 | 1 | | 1 | | 1 | |
| Q2; 17.78–28.44 | 7.09 [4.74–10.61] | <0.001 | 7.01 [4.69–10.50] | <0.001 | 5.92 [3.82–9.18] | <0.001 |
| Q3; 28.45–45.87 | 41.94 [28.556–61.57] | <0.001 | 41.93 [28.54–61.61] | <0.001 | 33.91 [22.21–51.79] | <0.001 |
| Q4; >45.87 | 204.40 [138.84–300.90] | <0.001 | 208.60 [141.51–307.51] | <0.001 | 147.94 [95.79–228.48] | <0.001 |
| **VAI** | | | | | | |
| Q1; <0.92 | 1 | | 1 | | 1 | |
| Q2; 0.92–1.47 | 3.96 [2.93–5.36] | <0.001 | 3.94 [2.91–5.33] | <0.001 | 2.76 [1.93–3.94] | <0.001 |
| Q3; 1.48–2.46 | 18.79 [14.18–24.89] | <0.001 | 19.05 [14.37–25.25] | <0.001 | 19.98 [14.34–27.84] | <0.001 |
| Q4; >2.46 | 97.52 [73.39–129.58] | <0.001 | 101.08 [75.96–134.52] | <0.001 | 127.08 [90.35–178.75] | <0.001 |
| **AIP** | | | | | | |
| Q1; <−0.149 | 1 | | 1 | | 1 | |
| Q2; −0.149–0.06 | 3.55 [2.67–4.72] | <0.001 | 3.89 [2.92–5.20] | <0.001 | 2.99 [2.12–4.22] | <0.001 |
| Q3; 0.06–0.28 | 15.77 [12.11–20.53] | <0.001 | 18.50 [14.08–24.31] | <0.001 | 20.19 [14.55–28.00] | <0.001 |
| Q4; >0.28 | 77.52 [59.38–101.19] | <0.001 | 97.07 [73.45–128.28] | <0.001 | 140.32 [99.82–197.26] | <0.001 |

**Note:**
Crude model: unadjusted; model 1: adjusted for age and sex; model 2; adjusted for model 1 plus FBS, SBP, DBP, BMI, and WC.

**Table 3 Multivariate logistic regression of different indices for hypertension.**

| Parameter | Crude model OR (95% CI) | p-value | Model 1 OR (95% CI) | p-value | Model 2 OR (95% CI) | p-value |
|---|---|---|---|---|---|---|
| **TyG index** | | | | | | |
| Q1; <8.35 | 1 | | 1 | | 1 | |
| Q2; 8.35−8.73 | 1.72 [1.51−1.95] | <0.001 | 1.58 [1.38−1.80] | <0.001 | 1.39 [1.22−1.59] | <0.001 |
| Q3; 8.74−9.16 | 2.34 [2.06−2.66] | <0.001 | 2.11 [1.85−2.40] | <0.001 | 1.67 [1.46−1.92] | <0.001 |
| Q4; >9.16 | 3.71 [3.25−4.23] | <0.001 | 3.27 [2.86−3.75] | <0.001 | 2.37 [2.04−2.76] | <0.001 |
| **TG/HDL** | | | | | | |
| Q1; <0.71 | 1 | | 1 | | 1 | |
| Q2; 0.71−1.14 | 1.34 [1.18−1.52] | <0.001 | 1.23 [1.08−1.40] | 0.002 | 1.02 [0.89−1.17] | 0.755 |
| Q3; 1.15−1.93 | 1.93 [1.70−2.19] | <0.001 | 1.73 [1.52−1.97] | <0.001 | 1.32 [1.15−1.51] | <0.001 |
| Q4; >1.93 | 2.46 [2.16−2.80] | <0.001 | 2.19 [1.92−2.51] | <0.001 | 1.53 [1.33−1.77] | <0.001 |
| **TyG-BMI** | | | | | | |
| Q1; <192.70 | 1 | | 1 | | 1 | |
| Q2;192.70−214.75 | 1.73 [1.52−1.96] | <0.001 | 1.57 [1.38−1.80] | <0.001 | 1.40 [1.21−1.63] | <0.001 |
| Q3; 214.76−239.42 | 2.53 [2.22−3.88] | <0.001 | 2.29 [2.01−2.62] | <0.001 | 1.87 [1.57−2.23] | <0.001 |
| Q4; >239.42 | 4.52 [3.96−5.17] | <0.001 | 4.16 [3.63−4.76] | <0.001 | 2.84 [2.22−3.63] | <0.001 |
| **TyG-WC** | | | | | | |
| Q1; <678.30 | 1 | | 1 | | 1 | |
| Q2; 678.30−733.92 | 1.74 [1.53−1.98] | <0.001 | 1.56 [1.36−1.78] | <0.001 | 1.39 [1.20−1.60] | <0.001 |
| Q3; 733.93−794.89 | 2.69 [2.37−3.07] | <0.001 | 2.35 [2.05−2.68] | <0.001 | 1.89 [1.61−2.22] | <0.001 |
| Q4; <794.89 | 4.25 [3.72−4.85] | <0.001 | 3.71 [3.23−4.25] | <0.001 | 2.69 [2.19−3.32] | <0.001 |
| **METS-IR** | | | | | | |
| Q1; <32.16 | 1 | | 1 | | 1 | |
| Q2; 32.16−36.69 | 1.68 [1.47−1.91] | <0.001 | 1.58 [1.39−1.80] | <0.001 | 1.22 [1.06−1.41] | 0.007 |
| Q3; 36.70−41.66 | 2.27 [2.00−2.59] | <0.001 | 2.06 [1.81−2.35] | <0.001 | 1.33 [1.13−1.57] | 0.001 |
| Q4; >41.66 | 3.71 [3.25−4.24] | <0.001 | 3.44 [3.00−3.93] | <0.001 | 1.59 [1.28−1.97] | <0.001 |
| **LAP** | | | | | | |
| Q1; <17.78 | 1 | | 1 | | 1 | |
| Q2; 17.78−28.44 | 1.55 [1.36−1.76] | <0.001 | 1.47 [1.29−1.68] | <0.001 | 1.25 [1.09−1.44] | 0.001 |
| Q3; 28.45−45.87 | 2.32 [2.04−2.64] | <0.001 | 2.17 [1.90−2.47] | <0.001 | 1.62 [1.40−1.87] | <0.001 |
| Q4; >45.87 | 3.44 [3.01−3.92] | <0.001 | 3.21 [2.81−3.67] | <0.001 | 2.11 [1.79−2.49] | <0.001 |
| **VAI** | | | | | | |
| Q1; <0.92 | 1 | | 1 | | 1 | |
| Q2; 0.92−1.47 | 1.25 [1.10−1.42] | 0.001 | 1.24 [1.09−1.41] | 0.001 | 1.04 [0.91−1.18] | 0.591 |
| Q3; 1.48−2.46 | 1.61 [1.42−1.83] | <0.001 | 1.59 [1.40−1.81] | <0.001 | 1.24 [1.09−1.42] | 0.001 |
| Q4; >2.46 | 2.22 [1.95−2.52] | <0.001 | 2.16 [1.89−2.45] | <0.001 | 1.54 [1.34−1.77] | <0.001 |
| **AIP** | | | | | | |
| Q1; <−0.149 | 1 | | 1 | | 1 | |
| Q2; −0.149−0.06 | 1.37 [1.20−1.55] | <0.001 | 1.26 [1.11−1.43] | 0.001 | 1.05 [0.92−1.20] | 0.495 |
| Q3; 0.06−0.28 | 1.96 [1.72−2.23] | <0.001 | 1.76 [1.54−2.00] | <0.001 | 1.34 [1.17−1.54] | <0.001 |
| Q4; >0.28 | 2.48 [2.18−2.83] | <0.001 | 2.22 [1.94−2.53] | <0.001 | 1.55 [1.35−1.79] | <0.001 |

**Note:**
Crude Model: unadjusted; model 1: adjusted for age and sex; model 2; adjusted for model 1 plus FBS, TC, TG, LDL-c, HDL-c, BMI, and WC.

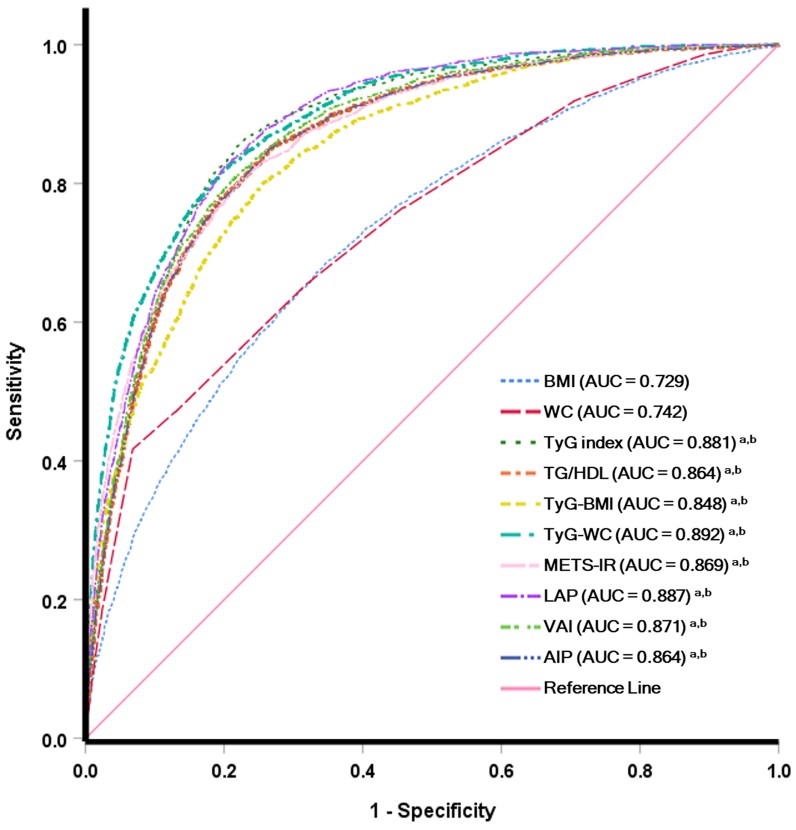

**Figure 2  Receiver operating characteristic analysis for predicting metabolic syndrome.**

for the LAP, TyG index, VAI, METS-IR, TG/HDL-c, AIP, and TyG-BMI were relatively high, with AUC ranging from 0.887 to 0.848. Additionally, the AUC values for IR surrogates were higher than traditional obesity indices such as WC and BMI (AUC values of 0.742 and 0.729, respectively). The difference in AUC between each IR marker is also shown in Fig. S1. Concerning the ability to predict hypertension, TyG-BMI had the highest AUC value of 0.659 (95% CI [0.648–0.669]), with a cut-off value of 211.54 and the Youden index of 0.236. Furthermore, TyG-WC, TyG index, METS-IR, and LAP had an AUC value (ranging from 0.655 to 0.634) higher than traditional obesity indices such as BMI (AUC: 0.630 (95% CI [0.619–0.640])) and WC (AUC: 0.618 (95%CI [0.607–0.629])). The difference in AUC between each IR marker is also shown in Fig. S2.

Insulin surrogate markers are more effective in predicting metabolic syndrome (MetS) in younger police personnel (age < 48 years), as well as hypertension, compared to older personnel, displayed in Tables S1 and S2.

Furthermore, this study combined two markers to predict the occurrence of MetS and hypertension. The combined IR markers took into consideration various factors used in calculating different IR marker formulas, such as TG, HDL-c, WC, FBS, and BMI, to ensure comprehensive coverage of all factors. The combined IR markers included TyG-BMI+TyG-WC, TyG-BMI+TG/HDL, TyG-BMI+LAP, TyG-BMI+VAI, TyG-BMI+AIP, TyG-WC+TG/HDL, TyG-WC+METS-IR, and TyG-WC+AIP. The study found that the

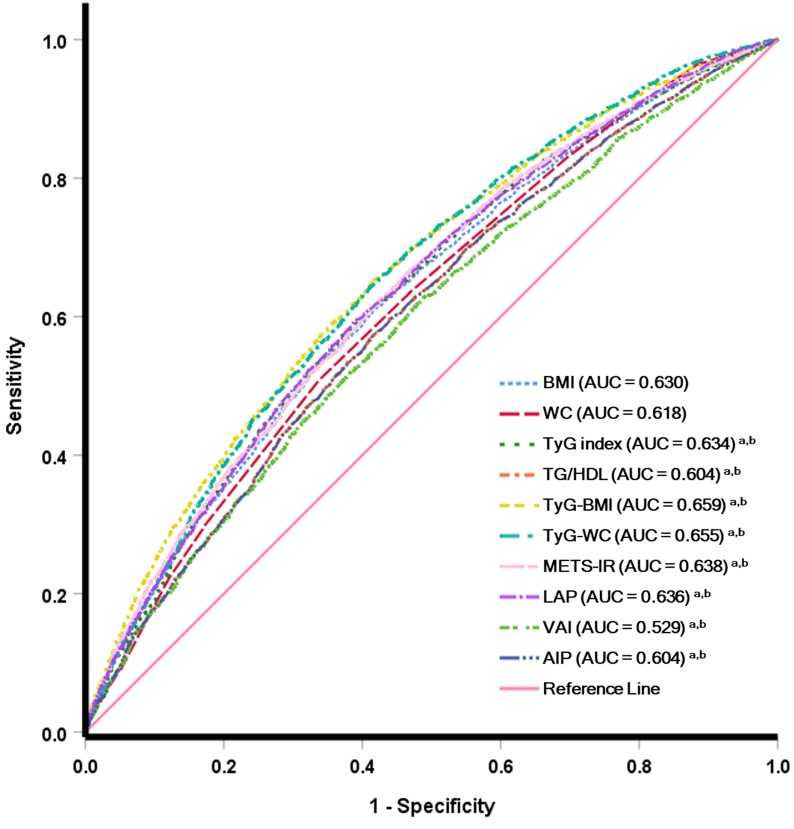

**Figure 3 Receiver operating characteristic analysis for predicting hypertension.**

combined IR markers provided better predictive results for MetS and hypertension than single IR markers. Predicting MetS, the combined IR marker TyG-WC+METS-IR (AUC 0.895, Yoden index 0.628) yielded better results than the single IR marker TyG-WC (AUC 0.892, Yoden index 0.620), as shown in Table S3. For predicting hypertension, the combined marker TyG-BMI+TyG-WC (AUC 0.661, Yoden index 0.241) provided better predictive performance compared to the best-performing single IR marker TyG-BMI (AUC 0.659, Yoden index 0.236), as shown in Table S4.

## DISCUSSION

Thai police officers perform various duties, including law enforcement, crime prevention, investigation, patrol, community policing, and traffic management. They also engage in administrative tasks, training, and community engagement efforts to promote public safety. The law enforcement officer is a high-stress vocation associated with higher cardiovascular disease prevalence and mortality risk (*Magnavita et al., 2018*). Police personnel reportedly have a high prevalence of hypertension and MetS (*Yates et al., 2021*), these may contribute to the health deterioration and unavailability of law enforcement employees.

A study involving 7,852 police officers found a higher prevalence of MetS at 30.8% compared to the general population in southern Thailand, where a survey conducted

**Table 4 Multivariate logistic regression of deference indices for hypertension.**

| IR surrogate index | AUC (95% CI) | *p*-value* | Sensitivity (%) | Specificity (%) | Cut-off | Youden index |
|---|---|---|---|---|---|---|
| **To predict metabolic syndrome** | | | | | | |
| BMI | 0.729 [0.719–0.739] | <0.001 | 68.86 | 65.05 | 24.76 | 0.339 |
| WC | 0.742 [0.732–0.751] | <0.001 | 41.76 | 93.13 | 89.00 | 0.349 |
| TyG index | 0.881 [0.873–0.888] | <0.001 | 85.38 | 78.14 | 8.88 | 0.635 |
| TG/HDL | 0.864 [0.856–0.872] | <0.001 | 84.68 | 73.71 | 1.29 | 0.584 |
| TyG-BMI | 0.848 [0.840–0.856] | <0.001 | 79.93 | 74.55 | 221.59 | 0.545 |
| TyG-WC | 0.892 [0.885–0.899] | <0.001 | 81.29 | 80.72 | 760.77 | 0.620 |
| METS-IR | 0.869 [0.862–0.877] | <0.001 | 80.75 | 77.32 | 38.45 | 0.581 |
| LAP | 0.887 [0.880–0.894] | <0.001 | 82.45 | 79.95 | 34.62 | 0.624 |
| VAI | 0.871 [0.863–0.878] | <0.001 | 81.50 | 77.87 | 1.78 | 0.594 |
| AIP | 0.864 [0.856–0.871] | <0.001 | 84.68 | 73.69 | 0.11 | 0.584 |
| **To predict hypertension** | | | | | | |
| BMI | 0.630 [0.619–0.640] | <0.001 | 53.68 | 65.76 | 24.80 | 0.194 |
| WC | 0.618 [0.607–0.629] | <0.001 | 50.74 | 66.36 | 84.00 | 0.171 |
| TyG | 0.634 [0.624–0.645] | <0.001 | 58.07 | 62.15 | 8.77 | 0.202 |
| TG/HDL | 0.604 [0.593–0.614] | <0.001 | 57.14 | 58.63 | 1.16 | 0.158 |
| TyG-BMI | 0.659 [0.648–0.669] | <0.001 | 65.25 | 58.34 | 211.54 | 0.236 |
| TyG-WC | 0.655 [0.644–0.665] | <0.001 | 64.45 | 58.73 | 727.47 | 0.232 |
| METS-IR | 0.638 [0.627–0.649] | <0.001 | 62.55 | 57.41 | 36.19 | 0.200 |
| LAP | 0.636 [0.626–0.647] | <0.001 | 59.36 | 61.26 | 28.81 | 0.206 |
| VAI | 0.529 [0.581–0.603] | <0.001 | 62.78 | 51.18 | 1.34 | 0.140 |
| AIP | 0.604 [0.593–0.614] | <0.001 | 56.87 | 58.86 | 0.06 | 0.157 |

**Note:**
* Null hypothesis, AUC = 0.5; BMI, body mass index; WC, waist circumference; TyG index, triglyceride glucose index; TG/HDL-c, triglycerides/high-density lipoprotein cholesterol ratio; TyG-BMI, TyG index with body mass index; TyG-WC, TyG index with waist circumference; METS-IR, metabolic score for insulin resistance; LAP, Lipid accumulation product; VAI, Visceral obesity index; AIP, atherogenic index of plasma.

between 2019–2020 reported a prevalence of 21.1% for MetS (*Aekplakorn, Puckcharern & Satheannoppakao, 2021*). Additionally, the prevalence of hypertension was 51.1% among police officers, which is higher than in general participants (21.1%) in the same region (*Aekplakorn, Puckcharern & Satheannoppakao, 2021*).

The most important observations from this research were that a strong relationship existed between the high prevalence of MetS and hypertension among police officers and eight IR surrogate markers. In addition, we found that LAP, METS-IR, TyG index, TyG-BMI and TyG-WC are good predictors for hypertension, at optimal cut-off better than traditional obesity indices such as BMI and WC. Among them, TyG-BMI had the best performance in predicting hypertension. Furthermore, comparing the predictive value of eight IR surrogates with prevalence MetS, TyG-WC demonstrated the greatest AUC in predicting MetS. In addition, AIP, LAP, METS-IR, TG/HDL-c, TyG index, TyG-BMI and VAI was superior to the traditional anthropometric index in predicting the presence of MetS. In addition, this study found that when two IR markers were combined for predicting MetS and hypertension, a significantly improved predictive value for disease risk was observed, as evidenced by higher AUC and Yoden index. Moreover, IR surrogate

markers demonstrate greater effectiveness in predicting both MetS and hypertension in younger police personnel (age < 48 years) compared to older officers.

Epidemiological research reveals that law enforcement workers have a higher risk of cardiovascular disease and mortality due to their high-stress profession and sedentary lifestyles. Police officers had high rates of hypertension, hyperlipidemia, and MetS (*Yates et al., 2021*). Therefore, research is needed to be considered as a significant indicator for predicting hypertension and MetS, which should help reduce the risk of hypertension and MetS in police personnel promptly. Insulin resistance has been found to play a significant causal role in developing hypertension (*Brosolo et al., 2022*) and cardiovascular diseases (*Di Pino & DeFronzo, 2019*). Endothelial dysfunction, vascular resistance, the activity of the sympathetic nervous system, renal sodium and fluid retention, and the subsequent renin-angiotensin-aldosterone system may play a crucial part in the etiology of hypertension when insulin resistance is present (*Brosolo et al., 2022*; *Janus et al., 2016*). The previous report indicated the correlation between the LAP, TG/HDL-c, TyG index and VAI with a HOMA-IR, which reflex the predictive ability of insulin resistance by these indices (*Huang et al., 2022*). Moreover, the TyG index, BMI, and WC combination suggested an increased ability to diagnose insulin resistance (*Er et al., 2016*). In addition, HOMA-IR was reported to correlate positively with SBP and DBP (*Quesada et al., 2021*).

Previous studies reveal the TG/HDL-c, TyG index TyG-BMI and TyG-WC potential for distinguished hypertension, and TyG-BMI and TyG-WC had a better ability than HOMA-IR (*Yuan, Sun & Kong, 2022*; *Zhang et al., 2021*). Interestingly, this study found that an indicator that combined the TyG index with BMI, TyG-BMI, had superior performance in predicting hypertension better than TyG index and BMI. Furthermore, TyG-WC had a higher predictive value for MetS than TyG index and WC. The fact that TyG-BMI and TyG-WC are more accurate predictors of MetS and hypertension than TyG index, WC and BMI is of clinical relevance and could be indicated that insulin-related lipid indices may be more accurate for predicting hypertension when taking body fat composition into consideration.

METS-IR, a simple insulin resistance index for the evaluation of cardiometabolic risk. In the present study, we found the superiority of METS-IR compared with traditional obesity indices, BMI and WC, in predicting hypertension and MetS. Previous studies revealed a high predictive value for the prevalence of MetS and hypertension, similar to our finding (*Bello-Chavolla et al., 2018b*; *Liu, Fan & Pan, 2019*; *Yuan, Sun & Kong, 2022*).

LAP is calculated by combining WC, an indication of abdominal obesity, and TG, which is associated with visceral obesity (*Kahn, 2005*). LAP and VIA, indicators of visceral adiposity and adipose tissue dysfunction, were found to be correlated with insulin resistance, hypertension and MetS (*Huang et al., 2022*; *Sung et al., 2020*). Furthermore, LAP and VAI have been widely explored for their association with the incidence and prevalence of type 2 diabetes, and it has been proven to be superior to traditional anthropometric indices in the prediction of type 2 diabetes mellitus and adverse cardiovascular events (*Ahn et al., 2019*; *Ramdas Nayak et al., 2020*). In this study, LAP and VAI presented a high predictive value for hypertension and MetS compared to the reference group. In addition, LAP revealed a better identification ability for hypertension

and MetS than traditional obesity indices such as BMI and WC. Furthermore, our findings indicate that VAI outperforms conventional obesity indices in predicting MetS, suggesting that relying solely on BMI or WC, which have limited accuracy in assessing subcutaneous fat accumulation, may be inadequate in accurately determining an individual's health status. On the other hand, taking WC, BMI and TG together into consideration, LAP and VAI could increase the sensitivity and specificity of hypertension and MetS prediction.

AIP is a new biomarker for predicting metabolic alterations related to cardiovascular disease (*Kahaer et al., 2022*; *Kammar-García et al., 2020*). In this study, the partial correlation analysis suggested that AIP significantly correlated with the prevalence of hypertension and MetS. The survey of young Mexican adults (aged 18–22 years) reported statistically significant for predicting hypertension and MetS with a high AUC value of 0.8 and 0.95, respectively (*Kammar-García et al., 2020*). Similar to the longitudinal research in Taiwanese adults over 40 years, the association between AIP and hypertension and MetS was reported; however, the relationship with hypertension disappeared after age 65 (*Li et al., 2021*). This study's relationship between AIP and MetS provides similar findings; however, the relationship between AIP and hypertension gives distinct outcomes. AIP demonstrated a better capacity to predict MetS than conventional anthropometric indices but an inferior ability to predict hypertension in all participants. Interestingly, when we looked at young police aged < 48 years, we found that AIP had better predictive capabilities for hypertension than older age groups. Our study revealed that the combination of lipid indicators represented as AIP might predict hypertension in young people.

This study investigated the association between the IR index and hypertension in various age groups. The IR indexes have more ability to predict hypertension in younger police officers than in older officers. Similar outcomes were observed for the predictive value of each IR surrogate in MetS, indicating that the combination of lipid and obese indices can predict MetS and hypertension in young police officers. Our findings suggested better clinical relevance for the younger population, especially in law enforcement personnel, which may lower the burden of developing cardiovascular diseases caused by hypertension and metabolic diseases in later life.

Our study had some limitations to be discussed. First, because the study was cross-sectional, it could not determine the causal association between surrogate IR indicators and the risk of hypertension and MetS. Second, the lacking participants' data, such as mental health, sedentary behavior, alcohol consumption and smoking status, could not further determine the influence of these factors on the outcomes. Third, this study was not conducted HOMA-IR, the gold standard of insulin sensitivity evaluation, and was used as an alternative tool for identifying insulin resistance. Finally, the study population consisted of police personnel only, and most police officers were men. This may result in an inaccurate assessment of the results, and we will consider redesigning our research in the future to address this deficiency.

## CONCLUSIONS

The study found that Thai police officers had a high prevalence of MetS and hypertension. These conditions were significantly associated with eight IR surrogate markers, including

AIP, LAP, METS-IR, TG/HDL-c, TyG index, TyG-BMI, TyG-WC, and VAI. All eight IR surrogates were found to be good predictors for hypertension and MetS in police officers. Among them, the TyG-BMI index performed the best in predicting hypertension, while the TyG-WC index performed the best in predicting MetS. Notably, the IR indices were particularly effective in predicting MetS in younger police personnel.

## ACKNOWLEDGEMENTS

The authors are grateful for the technical support from the Center of Excellence Research of Melioidosis and Microorganisms (CERMM), and the Walailak University Medical Technology Clinic (WU-MeT), Walailak University.

### Funding
This research was funded by the Medical Technology Clinic (WU-MeT), Walailak University, Thailand. The funders had no role in study design, data collection and analysis, decision to publish, or preparation of the manuscript.

### Grant Disclosures
The following grant information was disclosed by the authors:
Medical Technology Clinic (WU-MeT), Walailak University, Thailand.

### Competing Interests
The authors declare that they have no competing interests.

### Author Contributions
- Rujikorn Rattanatham performed the experiments, analyzed the data, prepared figures and/or tables, authored or reviewed drafts of the article, and approved the final draft.
- Jitbanjong Tangpong performed the experiments, prepared figures and/or tables, and approved the final draft.
- Moragot Chatatikun performed the experiments, prepared figures and/or tables, and approved the final draft.
- Dali Sun analyzed the data, prepared figures and/or tables, and approved the final draft.
- Fumitaka Kawakami analyzed the data, prepared figures and/or tables, and approved the final draft.
- Motoki Imai analyzed the data, prepared figures and/or tables, and approved the final draft.
- Wiyada Kwanhian Klangbud conceived and designed the experiments, performed the experiments, analyzed the data, prepared figures and/or tables, authored or reviewed drafts of the article, and approved the final draft.

### Ethics
The following information was supplied relating to ethical approvals (*i.e.*, approving body and any reference numbers):

The Walailak University Ethics Committee for Human Research approved this study (approval no. WUEC-21-349-01).

## Data Availability

The data is available at Zenodo: Wiyada Kwanhian Klangbud. (2023). Data and Code [Data set]. In Peerj "Assessment of eight insulin resistance surrogate indexes for predicting metabolic syndrome and hypertension in Thai law enforcement officers". Zenodo. https://doi.org/10.5281/zenodo.7890384

The raw data are available in the Supplemental Files.

## Supplemental Information

Supplemental information for this article can be found online at http://dx.doi.org/10.7717/peerj.15463#supplemental-information.

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
