# Peer review of "Assessment of eight insulin resistance surrogate indexes for predicting metabolic syndrome and hypertension in Thai law enforcement officers"

_PeerJ, doi:10.7717/peerj.15463_

## Round 0.1 · original submission · Minor Revisions

The reviews are in, and all are positive. Some minor issues have been identified which I'd ask you to address in a revision please; include clear explanation of changes in your cover letter detailing your response.

Reviewer 1 ·

Basic reporting

1. I highly suggested the authors replace some tables with corresponding figures to make it easier for readers. For example, a heatmap can be used for Table 2.

2. Please share your code on GitHub to make the analysis reproducible.

Experimental design

1. Table 5 has p-values for AUC but I'm not sure what test has been done on what metrics.

2. It's good that single surrogate markers can classify MetS and hypertension. However, will combining any of them improve the model? I highly suggested the authors test cases of multiple markers.

3. I highly suggested adding a heatmap of the correlation between surrogate markers to show the data distribution.

Validity of the findings

no comment

Reviewer 2 ·

Basic reporting

The study does not clearly explain the significance of only focusing law enforcement officers.

What is the relationship between the type of work that officers do and the primary outcomes of the research project.

The introduction and the discussion sections have not drawn the links the chosen study population and the outcomes.

Experimental design

What was the selection method used?
The authors need to demonstrate how they arrived to sample size of 7852 participants.
The authors report results of blood pressure, lipid profiles and glucose. However, in the methodology sections it is not clear how these parameters were measured.

Validity of the findings

No comment.

Reviewer 3 ·

Basic reporting

no comment

Experimental design

no comment

Validity of the findings

no comment

Additional comments

This study focuses on finding the best Insulin resistance surrogate indices in an occupational law enforcement officers from Thailand. This is an important study as there exists a high prevalence of metabolic syndrome and hypertension among Thai police officers. The work has been carefully designed and well-executed. However I think the conclusions are not precise as what indices is the best for predicting these metabolic syndrome or hypertension. I recommend major revisions based on my comments below.

Here are my comments:

Figure 1. For predicting metabolic syndrome. While there are many other indices that are almost similar, the authors claims TyG-WC is best in the conclusions. How different is TyG-WC from others, what is the p-value?

Also the same for the Figure 2. There are other indices similar to TyG-BMI, and one cannot claim TyG-BMI is the best.

Figures 1 and 2: Slightly thicker lines should be used for the curves and legends. Currently it is hard to read.

The reviewer would have enjoyed more if the manuscript is written well. The writing lags in many places and it definitely need major rewriting for the readers to understand well.

Here are some examples but NOT a complete list:
Abstract should be made interesting to reviewers and readers.
Line 39: “The prevalence of hypertension and MetS were 51.1% and 30.8%” should add a word “respectively” at the end.

Table 3 title: Multivariate logistic regression of deference indices for metabolic syndrome. Did authors meant different indices?

Conclusions should be rewritten: For example the sentence below doesn’t read well.
The TyG-BMI index had the best performance in predicting hypertension, TyG-WC index had the best performance in predicting MetS.

---

## Round 0.2 · accepted · Accept

Thank you for responding thoroughly to the issues raised. I am satisfied with these responses.